# Effect of Addition of a Mixture of Ethyl Esters of Polyunsaturated Fatty Acid of Linseed Oil to Liquid Feed on Performance and Health of Dairy Calves

**DOI:** 10.3390/ani14071048

**Published:** 2024-03-29

**Authors:** Mohammed K. Baba, Jadwiga Flaga, Zygmunt M. Kowalski

**Affiliations:** 1Department of Animal Nutrition, Biotechnology and Fisheries, University of Agriculture in Krakow, 30059 Krakow, Poland; bmohamm@nsuk.edu.ng (M.K.B.); j.flaga@urk.edu.pl (J.F.); 2Department of Animal Science, Faculty of Agriculture, Shabu-Lafia Campus, Nasarawa State University Keffi, Keffi 911019, Nigeria

**Keywords:** animal nutrition, immunity, growth, cattle, omega-3 fatty acids

## Abstract

**Simple Summary:**

Although there has been significant progress in management practices for the rearing of dairy calves in many countries in the last two decades, still the mortality and morbidity rates are unacceptably high. This study aimed to determine the effect of supplementing liquid feeds with a mixture of ethyl esters of polyunsaturated fatty acid of linseed oil (EEPUFA) on feed intake, body weight gain, feed efficiency, and health of dairy calves. We found better starter intake, dry matter intake, daily body weight gain, and a lower percentage of days with diarrhea relative to the number of days receiving treatment in the EEPUFA group than the control group. Overall, there is quite promising impact of EEPUFA on the growth performance and health of dairy calves.

**Abstract:**

This study aimed to determine the effect of supplementing liquid feeds with a mixture of ethyl esters of polyunsaturated fatty acid of linseed oil (EEPUFA; α-linolenic acid—64.5%, linoleic acid—16.1%, and oleic acid—19.4%) on feed intake, body weight gain, feed efficiency, and health of dairy calves. Thirty-six healthy female Holstein–Friesian calves (7 d of age, 41.2 ± 4.0 kg) were assigned to one of two treatment groups (18 calves per group), i.e., control or EEPUFA, and fed liquid feed (whole milk (WM) or milk replacer (MR)) either without or with 10 mL/d of EEPUFA supplementation, respectively, for 56 days (till 63 d of age). Average daily intake of WM and MR was similar between treatments (*p* = 0.94). Average daily total DM intake and average daily starter feed DM intake were higher for the EEPUFA group (*p* = 0.05 and *p* = 0.01, respectively). The average daily body weight gain was also higher for the EEPUFA group (55 g/d; *p* = 0.03), although final body weight turned out not to be significantly different between groups (75.6 kg vs. 79.0 kg, control vs. EEPUFA, respectively; *p* = 0.20). Supplementation of liquid feeds with EEPUFA did not affect feed efficiency (*p* = 0.37) and most of investigated health parameters. However, the percentage of days with diarrhea relative to the number of days receiving treatment was higher in the control group than the EEPUFA group (76 vs. 42, respectively; *p* = 0.04). Although the results of this preliminary study are promising, further research is needed to establish the dose effect of EEPUFA on the performance and health of calves.

## 1. Introduction

Future milk production of the dairy herd can be improved by the effective management of calves considering their growth rate [1,2,3] and health [4]. Calf mortality and morbidity are critical issues not only for future replacement [5] but also for producers and consumers since the high rates of mortality and morbidity lead to huge economic losses and animal welfare concerns [6]. Although there has been significant progress in management practices in the rearing of dairy calves in many countries over the last two decades, still the mortality and morbidity rates are unacceptably high [7,8]. In this context, any health improvement that promotes a better growth rate within the rearing period might be appreciated. As calves are subjected to infectious diseases, such as pneumonia, as well as to stressful conditions, stimulation of immune and antioxidative systems is required [9,10].

The use of polyunsaturated fatty acids (PUFAs) of plant or fish origin in milk replacers (MRs) for calves to stimulate their immune system functioning has received special attention as it resulted in better production parameters and improved immune responses [11,12,13,14,15]. Adding PUFA-containing oils to MRs enhanced body growth and alleviated symptoms of diarrhea and inflammatory disorders induced by viral or bacterial infections [15]. Melendez et al. [14] found a higher tendency for diarrhea in calves fed MR supplemented with canola oil than those supplemented with fish oil, rich in omega-3 PUFA. Omega-3 PUFAs, as opposed to omega-6 ones, have protective functions, preventing inflammatory conditions by lowering the expression of pro-inflammatory genes [16].

Linseed (*Linum usitatissimum* L.; flaxseed) and linseed oil have been traditionally used as nutritional supplements in both human and animal nutrition, especially as a source of α-linolenic acid (C18:3 omega-3; ALA), with a relatively low ratio of omega-6 to omega-3 fatty acids [17,18]. To improve the growth and immunity of calves, either flaxseed oil [12] or a mixture of fish and flaxseed in MR [11,19] or calf starter [20] as a source of omega-3 fatty acids were used.

Ethyl esters of PUFA of linseed oil (EEPUFA) might serve as a feasible alternative to the use of linseed oil. EEPUFAs have greater bioavailability than traditional triglycerides and are easier absorbed and integrated into various lipid fractions of the blood, owing to their simple molecular structure and more effective kinetics of free acid release, guaranteeing more rapid digestion [21,22]. EEPUFAs as compared to linseed oil are substantially less vulnerable to some biological processes such as oxidation, peroxidation as well as epoxidation [23,24]. EEPUFAs are in the liquid form and are more easily dissolved in MR than oils [25].

Śpitalniak-Bajerska et al. [26] found that adding EEPUFA to MR for newborn calves increased PUFA supply to the body and served as an antioxidant source. Although this study showed the positive effect of EEPUFA on the performance and health of calves, these effects were confounded using lyophilized apples together with EEPUFA. Moreover, the number of animals (nine per treatment) might have limited the power of that experiment.

We hypothesized that the addition of a mixture of EEPUFA has a positive effect on the performance and health of the preweaning dairy calves. This study aimed to determine the effect of supplementing liquid feeds with a mixture of EEPUFA on feed intake, body weight gain, feed efficiency, and health of dairy calves.

## 2. Materials and Methods

### 2.1. Ethical Approval

The experiment did not require acceptance from the local ethics commission because there were no painful procedures. The procedure of blood collection to assess the serum total protein content is routinely carried out for all calves born on the commercial farm selected for the experiment (Wojnowice Farm, Agromax Sp. z.o.o., Racibórz, Poland) in order to ensure proper colostrum feeding. The authors only used the results of this assessment to ensure that the calf selected for the experiment was healthy and properly protected.

### 2.2. Animals and Experimental Design

Thirty-six healthy female Holstein–Friesian calves (7 d of age, 41.2 ± 4.0 kg) were assigned to one of two treatment groups (18 calves per group), i.e., control or EEPUFA, and fed whole milk (WM) or MR either without or with EEPUFA, respectively, for 56 days (till 63 d of age). EEPUFA (LeenVit Group Sp. z o.o., Jaworzno, Poland) was a mixture of α-linolenic acid (64.5%), linoleic acid, (16.1%), and oleic acid (19.4%). Before their recruitment into the experiment, the calves were subjected to the standard farm procedures of colostrum administration and housing. They were kept indoors in their individual pens bedded with straw. As a part of a standard procedure performed on the farm, blood samples were collected from the jugular vein of calves (at 3 or 4 d of age) to assess the serum protein level. Only those animals with a blood serum total protein content higher than 5.2 g/dL were accepted for the experiment. Moreover, only healthy animals were selected for the experiment as well as those that were healthy from birth to 7 d of age. Dam parity, calving difficulties, birth weight, and weight at 7 d of age were also considered in the recruitment of animals and assigning them to experimental groups. Birth weight, parity (1—primiparous, 2—multiparous), number of parities, serum protein, and calving difficulty (0 = dam calved without difficulty; 1 = dam calved with difficulty) for control and EEPUFA groups were as follows: 41.11 ± 4.82 and 41.92 ± 4.63, 1.50 ± 0.51 and 1.50 ± 0.51, 1.67 ± 0.77 and 1.78 ± 0.94, 5.96 ± 0.58 and 6.07 ± 0.55, and 0.72 ± 0.46 and 0.50 ± 0.51, respectively.

All calves received WM till 14 d of age and then the regular MR (IgluVital; Josera GmbH & Co. KG, Kleinheubach, Germany) and ad libitum starter feed (SF) in the form of the so-called ‘dry TMR’ (Calf starter, Cargill, Warsaw, Poland). The chemical compositions of MR and SF, including fatty acids, are presented in Table 1.

The calves were served 2 L of liquid feed 3 times a day until they reached 12 d of life and then were fed 3 L of liquid feed 2 times a day until 63 d of age, which was the last day of the experiment. MR (96.3% DM, 20.8% CP, and 17.7% EE in DM; Table 1) was fed in an amount equal to 840 g of MR powder/d (140 g of MR powder in 1 L) dissolved in warm water (39 °C). The amount of daily WM and MR leftovers was measured.

A total of 10 mL/d of EEPUFA measured with an automatic pipette was poured directly into the bucket with morning portion of WM or MR and mixed thoroughly. Such a dose was established from the dosing of the EEPUFA mixture in humans as well as based on the previous study by Śpitalniak-Bajerska et al. [26]. 

The calves were given a new portion of SF every day, and the SF left by each calf was weighed every morning. From the first d of experiment, the calves were offered 500 g of SF/d, and this amount was increased by additional 500 g when the amount of refusals was less than 200 g.

### 2.3. Data Collection 

Once a week, samples of MR and SF were collected, pooled by month of the study, and subjected for chemical analysis. Samples were then dried at 55 °C for 72 h and grounded to pass through a 1 mm screen and analyzed for DM (No. 934.01), ash (No. 942.05), and CP (No. 976.05) according to AOAC [27]. Crude fat was determined according to AOAC [28] procedure number 920.39. 

The fatty acid (FA) compositions of MR and SF were determined by extracting their fat following the Folch et al. [29] technique as in Flaga et al. [13]. The samples were homogenized in the mixture of chloroform and methanol 2:1 *v*/*v* and then filtered. Methyl esters of FAs were prepared through the transmethylation of fat samples, adhering to the AOCS Official Method Ce 2–66 [30]. Gas chromatography (GC), as detailed by Rutkowska et al. [31], was employed for the separation of methyl esters. The Supelco 37 No. 47885-U standard (Sigma-Aldrich, St. Louis, MO, USA) was used to determine the recovery rates and correction factors for individual FA in lipid extract samples, as well as for FA identification and recovery of their methyl esters. The retention times of unsaturated methyl esters of FA (C18:2 linoleic, C18:3 α-linolenic, C18:3 γ-linolenic, C20:5, and C22:6) were confirmed by those recorded for natural plant oils (rapeseed, sunflower, evening primrose) and for fat extracted from fish (salmon), for which those acids are characteristic. The content of individual FA was expressed as % of total fatty acids. To ensure the robustness and reproducibility of the results, the analysis of each sample was performed in triplicates. 

Feed intake, body weight, and health were monitored and recorded individually. Feed intake was measured daily, body weight was measured by weighing the calves on 7, 21, 35, 49, and 63 d of life, and body weight gain was then calculated. The daily DM intakes were combined to obtain the average per week. Feed efficiency was determined by calculating the ratio of daily body weight gain (g/d) to daily DM intake (g/d). The health of the calves was monitored, and daily assessments of feces using a scale of 1 to 4 were performed, where 1 is normal (but not hard), 2 is soft (does not hold its form, resembles melting ice cream), 3 is liquid (runny, melts easily), and 4 is watery [32]. Health problems such as days in the trial when the calves were sick, the number of days receiving treatment, days with diarrhea (fecal fluidity ≥ 3 points), and the percentage of days with diarrhea relative to the number of days receiving treatment were recorded. Moreover, for each calf, the percentage of days with diarrhea relative to the number of days receiving treatment was calculated by dividing the days with diarrhea by the number of days receiving treatment, and the result was multiplied by 100 to obtain percentage.

### 2.4. Statistical Analysis

The data were cleaned and checked for outliers using Cook’s distance and standardized residual error method. The normality was tested using Shapiro–Wilk’s test, and the homogeneity of variance was tested by exploring the quantile–quantile plot and box plot in Proc Univariate of SAS [33]. The data on SF intake and total DM intake failed the assumptions of normality. Therefore, they were log-transformed before analysis.

The statistical analysis of performance data and fecal scores was performed using Proc Glimmix of SAS [33]. The best covariance structure was chosen based on having a lower value of Akaike information criteria. The effect of group (control vs. EEPUFA), time (14, 21, 28, 35, 42, 49, 56, and 63 d of age for the SF, total DM intakes, and fecal scores; 7, 21, 35, 49, and 63 d of age for data on body weight; 21, 35, 49, and 63 d of age for that of average daily weight gain and feed efficiency), and interaction of group and time were included as fixed effects while serum protein and dam parity (primi- vs. multiparous) were included as covariates. Other covariates (initial body weight, birth weight, weight at 7 d of age, calving difficulty, and parity number) were not included because they had *p*-values greater than 0.20. The random statements of time to account for the repeated effect of time were included in the model. Other data such as initial weight, final weight, and total body weight gain were analyzed without the effect of time.

The health problems, i.e., days in the trial when the calves were sick, number of days receiving treatment, days with diarrhea, and the percentage of days with diarrhea relative to the number of days receiving treatment, were analyzed using the non-parametric Mann–Whitney sum of rank test. 

The least squares mean (LSM) and standard errors of mean (SEM) were reported in tables and graphs. However, the LSM of the transformed variables were later back-transformed by taking the exponent of the LSM. Therefore, we reported the back-transformed LSM with their original log-transformed SEM values. The health parameters were reported as means and SEM except when stated otherwise. Statistical significance was declared at *p* ≤ 0.05 and tendencies at 0.05 < *p* ≤ 0.10.

## 3. Results and Discussion

Due to the well-documented beneficial effects of n-3 fatty acids [11,12,13,14,15,16], their supplementation in the diet of dairy calves is justified and may contribute to improving the well-being of calves in the critical first months of life. In this study, calves were fed liquid feed (WM or MR) without or with the addition of EEPUFA in order to determine the effect of such supplementation on feed intake, body weight gain, feed efficiency, and health of dairy calves.

### 3.1. Effects of Supplementation of Liquid Feed with EEPUFA on Performance of Dairy Calves

Average daily liquid feed DM intake was similar between treatments (*p* = 0.94; Table 2).

On the other hand, calves in the EEPUFA group consumed 39 g/d more SF than calves from the control group (*p* < 0.05). The average daily total DM intake was 59 g/d higher (*p* < 0.01) in the EEPUFA group than in the control group. The SF and total DM intake were both affected by group and time (Figure 1a,b). 

However, there was no interaction between group and time on any parameter of intake studied. Higher DM intake may promote better growth and development of postnatal calves. Our results are similar to that obtained by Śpitalniak-Bajerska et al. [26]. Although in their study, higher starter intake in calves that consumed MR supplemented with EEPUFA was confounded by the use of lyophilized apples. The improved DM intake in EEPUFA compared to the control group in our study could be related to the immunomodulatory effects of EEPUFA, which may increase immune and antioxidative capacity and prevent diseases, thereby promoting better health and appetite in the calves. Enhanced immunity and health, and in turn better growth of calves supplemented with EEPUFA, may be caused by the good absorption of fatty acids from ethyl esters of PUFA in the small intestines [21,22]. In the study of Śpitalniak-Bajerska et al. [26], supplementation of MR with EEPUFA and lyophilized apples lowered blood TNF-α concentration as well as total antioxidant capacity, concentration of malondialdehyde, and activity of glutathione peroxidase when compared to the calves from the control, not the supplemented group. Such positive anti-inflammatory effects may be related to the presence of omega-3 FA in the EEPUFA since its main FA is ALA (64.5% of total FA), belonging to the omega-3 family. The omega-3 fatty acids have well-documented anti-inflammatory effects. Also, in calves [14,26,34], better total DM intake (MR + starter) was found in the calves fed MR supplemented with fish oil, containing omega-3 FA, than the canola oil containing more omega-6 and omega-9 FAs. In our earlier study [13], supplementation of MR by DHA-rich algae (DHA, docosahexaenoic acid) improved performance and induced beneficial effects on immunological variables in rearing calves; mRNA expression of IL-1β, TNF-α, and nuclear factor-κB subunit decreased linearly with increasing doses of DHA-rich algae. On the other hand, MR and starter intake linearly decreased with increasing doses of DHA-rich algae, perhaps due to the low palatability of the additive. In the present study, we did not observe any signs of the low palatability of the EEPUFA. The animals willingly consumed the entire dose and there were no refusals, which proves the high palatability of the feed. Additional evidence is the lack of differences in liquid feed consumption between groups (*p* = 0.94). Although the immunological responses were not determined in this study, we may assume that higher SF and total DM intake could be explained by the effect of FAs, especially ALA, on the immunological responses. 

There was no difference in the initial body weight between animals from both groups (Table 2). The body weight of calves was higher in the EEPUFA group (*p* = 0.04; Figure 2a). 

At 63 d of age, the control group calves were 3 kg lighter than the EEPUFA calves, but the differences between treatments were only numerical (*p* = 0.20; Table 2). On the other hand, there was a tendency (*p* = 0.10) for increased total body weight gain in the EEPUFA group. Moreover, the average daily body weight gain of supplemented calves was 55 g/d higher than in the control group (*p* = 0.03; Table 2 and Figure 2b).

All the parameters of body weight were affected by time (Figure 2a,b). However, there was no interaction between group and time effects on any parameter studied. These results indicate that the supplementation of MR by EEPUFA supported better growth of calves, similar to Śpitalniak-Bajerska et al. [26]. However, the average daily body weight gain recorded in this study (617 and 672 g/d in the control and EEPUFA group, respectively) was higher than in Śpitalniak-Bajerska et al. [26] where EEPUFA-supplemented calves grew 593 g/d. The reason for these differences may have been caused by the fact that our calves grew up from 7 to 63 d of age, but in the study of Śpitalniak-Bajerska et al. [26], the growth was measured between 14 and 42 d of age. Different linseed-based sources of fatty acid improved the growth rate of calves in other studies [12,14,15,20], and this effect could also be linked to the consumption of PUFAs, especially omega-3 ones. Their anti-inflammatory effect supported better intake, which in turn provided more nutrients for faster growth. 

As there are no differences between groups in feed efficiency (Table 2 and Figure 2c), better growth of the EEPUFA calves was not caused by positive changes in their metabolism [35]. On the other hand, as stated earlier, the higher daily body weight gain could be attributed to higher total DM intake. In our previous study [13], increased feed efficiency was observed when DHA-containing algae was added to MR. However, this effect was observed in the early life of calves (first month) and decreased in the next weeks. Also, Karcher et al. [12] observed an increased feed efficiency in calves fed flax oil as a source of fatty acid in MR compared with fish oil or the control group. These discrepancies in feed efficiency may be related to the different PUFA compositions of the additives used.

On the other hand, increased dietary PUFA may not necessarily translate into enhanced animal performance but can still be effective when it comes to health. In a study of Fusaro et al. [36] feeding ewes with a diet enriched with extruded linseed increased the amount of PUFA in their milk and at the same time decreasing omega-6 to omega-3 ratio. Feeding lambs with such milk did not affect their performance but changed their meat composition, specifically it contained a higher share of PUFA (especially omega-3). This is in accordance with other studies where a greater dietary PUFA supply led not only to their increased proportions in blood but also in many tissue compartments (especially cell membranes) where they affect local inflammatory processes and thus support immunity [37].

### 3.2. Effect of Ethyl Esters of Polyunsaturated Fatty Acid (EEPUFA) Added to Milk or Milk Replacer on the Health of Dairy Calves

Health parameters, such as fecal fluidity, days in the trial when the calves were sick, number of days receiving treatment, and the average days with diarrhea, did not differ between the groups in any of the study periods (Table 3; Figure 3). However, they were numerically more favorable in the EEPUFA group. 

Moreover, the percentage of days with diarrhea relative to the number of days receiving treatment was higher in the control group than the EEPUFA group (76 vs. 42, respectively; *p* = 0.04). It indicates that despite having close to similar health challenges, the percentage of days spent treating animals from the control group because of the higher incidence of diarrhea was greater than in the EEPUFA group. These findings are similar to those reported in other studies [13,15,25]. Hill et al. [15] observed lowered symptoms of diarrhea and diseases in calves when their MR was supplemented by a blend of butyric acid, coconut oil, and flax oil. Flaga et al. [13] showed reduced fecal score in the dairy calves fed MR supplemented with DHA-rich algae, and according to the authors of this study, the improved immune system and health of the calves were attributed to the supplementation of DHA. Also, Śpitalniak-Bajerska et al. [26] showed lower fecal scores when calves were fed MR supplemented with EEPUFA and lyophilized apples. Although we did not analyze the immune and oxidative status of calves, which was a weakness of our study, based on the literature, we may conclude that better health of calves from the EEPUFA group could result from the above-discussed anti-inflammatory and antioxidative effects of PUFA, especially ALA. The anti-inflammatory properties of omega-3 FA, including ALA, may help in the mitigation of inflammation by reducing the activity of signaling molecules, such as cytokines, and the expression of pro-inflammatory genes [16]. 

Supplementing milk or MR with 10 mL/d of the mixture of EEPUFA had a positive effect on the performance of calves by improving DM intakes and body weight gains. Further studies are needed to establish the possible dose effect of EEPUFA. Moreover, the most effective stage of the calf’s age for such supplementation should also be explored together with the possible long-term benefits of EEPUFA beyond the preweaning stage in dairy calves.

## 4. Conclusions

Our study contributes to the existing literature data on beneficial effects of PUFA dietary supplementation. It seems that any increase in their proportion in the diet through various sources produces some beneficial effects and should be recommended and introduced into practice by veterinarians, technicians, and farmers. However, EEPUFA may have some advantages over the other sources of PUFAs due to their greater bioavailability and stability and the possibility of influencing the composition of their mixture. Moreover, as shown in this experiment, the addition of EEPUFA had no effect on feed intake through decreased palatability, which is a common phenomenon observed when using, for example, fish oil or algae. Although their detailed impact on the immune system and its functions remains the subject of our further research, based on the results obtained, we can conclude that the addition of 10 mL/d of EEPUFA to the WM and then to MR of rearing calves already proves to enhance their performance and health. The improvement in DM intake but not feed efficiency as well as promoting better health and probably better immunity were responsible for better growth of calves.

## Figures and Tables

**Figure 1 animals-14-01048-f001:**
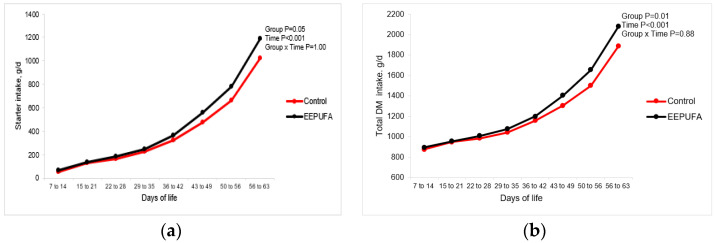
Daily dry matter intake of starter feed ((**a**) g/d) and total daily dry matter intake ((**b**) g/d) in dairy calves fed milk or milk replacer supplemented with 0 (control) or 10 mL/d of EEPUFA. The figures present LSM obtained by the model with repeated measures having fixed effects of group and time and their interaction.

**Figure 2 animals-14-01048-f002:**
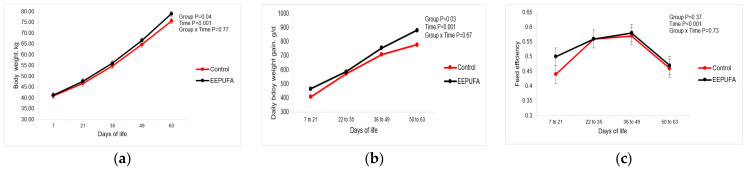
Body weight ((**a**) kg), daily body weight gain ((**b**) g/d), and feed efficiency (**c**) in dairy calves fed milk or milk replacer supplemented with 0 (control) or 10 mL/d of EEPUFA. The figures present LSM obtained by the model with repeated measures having fixed effects of group and time and their interaction.

**Figure 3 animals-14-01048-f003:**
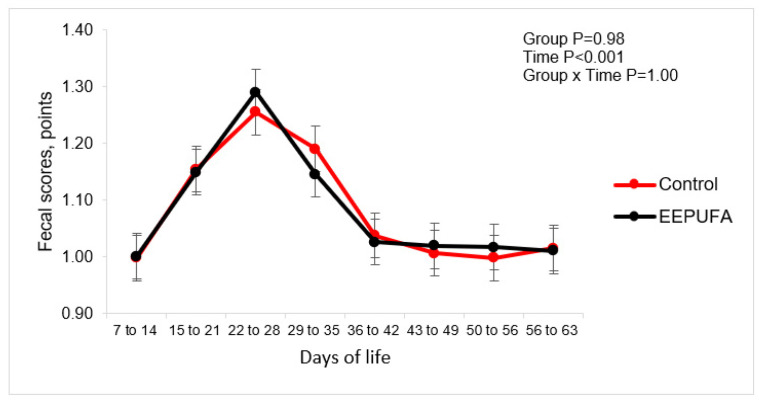
Fecal scores in dairy calves fed milk or milk replacer supplemented with 0 (control) or 10 mL/d of EEPUFA. The figures present LSM obtained by the model with repeated measures having fixed effects of group and time and their interaction.

**Table 1 animals-14-01048-t001:** Chemical compositions of milk replacer and starter feed.

Items	Milk Replacer ^1^	Starter Feed ^2^
Dry matter, %	96.34	90.34
Ash, % DM	7.91	7.47
Crude protein, % DM	20.82	20.14
Crude fat, % DM	17.70	5.19
Fatty acids, % of total fatty acids		
C4:0	0.08	
C8:0	0.80	
C10:0	1.08	0.07
C10:1 n-2	0.06	
C12:0	8.85	0.58
Iso-C 13:0	0.24	
C12:1	0.04	
C14:0	6.17	2.67
C14:1 n-5	0.06	0.08
C15:0	0.27	
C16:0	36.25	23.31
C16:1 n-7	0.76	1.26
C17:0	0.06	0.16
C18:0	5.13	5.68
C18:1 trans n-9	0.26	
C18:1 cis n-9	29.60	28.30
C18:2 n-6	8.54	31.89
C18:3 n-6		4.83
C20:0	0.13	0.14
C20:1 n-9	0.10	0.20
C18:3	0.83	
CLA cis-9 trans-11	0.02	
CLA cis-10 trans-12	0.03	
SUM	97.37	99.14
Unidentified	0.63	0.86

^1^ IgluVital; Josera GmbH & Co. KG, Kleinheubach, Germany; ^2^ dry TMR (Calf Starter, Cargill, Warsaw, Poland).

**Table 2 animals-14-01048-t002:** LSM of the effects of polyunsaturated fatty acid ethyl esters of linseed oil (EEPUFA) added to whole milk or milk replacer on daily starter feed DM intake (g/d), total DM intake (g/d), body weight (kg), daily body weight gain (g/d), and feed efficiency in dairy calves.

Items	Group		SEM ^3^	*p*-Value		
	Control	EEPUFA		Group	Time	Group × Time
Average daily liquid feed DM intake, g/d ^1^	808	808	0.0	0.94	<0.001	0.85
Average daily starter feed DM intake, g/d ^1^	266 ^b^	304 ^a^	0.1	0.05	<0.001	1.00
Average daily total DM intake, g/d ^1^	1176 ^b^	1235 ^a^	0.0	0.01	<0.001	0.88
Initial body weight, kg ^2^	41.4	41.0	1.00	0.80		
Final body weight, kg ^2^	75.6	79.0	1.90	0.20		
Total body weight gain, kg ^2^	34.6	37.7	1.29	0.10		
Average daily body weight gain, g/d ^1^	617 ^b^	672 ^a^	18.2	0.03	<0.0001	0.67
Feed efficiency ^1,4^	0.51	0.53	0.010	0.37	<0.0001	0.73

^1^ Calculated from the model based on the repeated effect of time; ^2^ analyzed without the repeated effect of time; ^3^ standard error of the mean; ^4^ calculated as average daily body weight gain g/d divided by average daily total DM intake g/d; and ^a,b^—means with different superscripts are significantly different.

**Table 3 animals-14-01048-t003:** Effect of ethyl esters of polyunsaturated fatty acid (EEPUFA) added to milk or milk replacer on the health of dairy calves.

Description	Group		SEM ^1^	*p*-Value		
	Control	EEPUFA		Group	Time	Group × Time
Fecal fluidity ^2,3,4^, point	1.1	1.1	0.04	0.98	<0.001	1.00
Total number of calves	18	18				
Number of calves sick (treated by a veterinarian)	10	7				
Days in the trial when the calves were sick ^5^	15	18	3.7	0.35	-	-
Number of days receiving treatment ^5^	4.0	3.3	1.0	0.38	-	-
Days with diarrhea ^5^	1.9	1.6	0.58	0.22	-	-
Percentage of days with diarrhea relative to the number of days receiving treatment	76	42	14.8	0.04	-	-

^1^ Standard error of the mean; ^2^ least squares mean; ^3^ data subjected to the analysis of variance for repeated measurements; ^4^ 1 = normal, 2 = soft, 3 = flowing, dripping, and 4 = watery [32]; and ^5^ means.

## Data Availability

Data is contained within the article.

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
