# Peer review of "Effect of Addition of a Mixture of Ethyl Esters of Polyunsaturated Fatty Acid of Linseed Oil to Liquid Feed on Performance and Health of Dairy Calves"

_animals, 2024, doi:10.3390/ani14071048_

Round 1

Reviewer 1 Report

Comments and Suggestions for Authors

The study, titled "Effect of Adding a Mixture of Ethyl Esters of Polyunsaturated Fatty Acids from Linseed Oil on the Performance and Health of Dairy Calves," aims to explore the impact of ethyl esters of polyunsaturated fatty acids from linseed oil on the performance and health of dairy calves. This research topic is highly relevant and intriguing, aligning well with the journal's scope. However, there are several areas that could benefit from improvement:

- I recommend rephrasing the abstract to enhance readability. Consider incorporating more numerical data along with significance levels to provide a clearer overview of the study's findings.

- Consider revising the keywords by removing those already present in the article's title. This will ensure a more diverse set of terms for indexing and searching.

- I suggest using a simpler abbreviation for the term "Ethyl Esters of PUFA (EEPUFA) of Linseed Oil." Furthermore, since the abbreviation has already been used in the abstract, repeating it in the text may be unnecessary.

- I disagree with the authors' assertion regarding the ethical committee, particularly considering that blood sampling was conducted, which is not a routine procedure in dairy cattle farming.

- The materials and methods section has some shortcomings that hinder the reproducibility of the research. Specifically, it is unclear how long the trial lasted in total. Additionally, I recommend providing more details about the dietary protocol used for food administration.

- Enhance the discussion by including the study's limitations and practical applications. This will provide a more comprehensive understanding of the research's implications and contribute to the broader scientific discourse

Author Response

The authors would like to thank the Reviewer for meaningful and insightful review of our manuscript. The details of our revisions/corrections as well as point-by-point response to the reviewer’s comments can be found in the attached file.

Reviewer 2 Report

Comments and Suggestions for Authors

The authors evaluate the effect of supplementing liquid feeds with a mixture of ethyl esters of polyunsaturated fatty acid of linseed oil (EEPUFALO) on feed intake, body weight gain, feed efficiency, and health of dairy calves. The results can be application and promotion. It is useful. The study is fit with the aim of the Journal, and it is meaningful.

Overall, it can be accepted after minor revision. Some of the details in the manuscript need to be improved.

In line 188, “Average daily DM intake of milk and MR were similar between treatments (Table 2).” However, I can not find the results about average daily DM intake of milk and MR in Table 2. Please check it.

In line 199 and Figure 1, the authors use “A” and “a”, delete the form of “A” and “B”.

In line 264, I think the vocabulary “Contrary” here is not very exactly.

In line 284, it is better to add the P value behind this result sentence.

In line 309, it is necessary to add the description of Figure 3 in the text.

Author Response

(The authors gave the same response as above.)

Reviewer 3 Report

Comments and Suggestions for Authors

The paper, titled " Effect of addition of a mixture of ethyl esters of 2 polyunsaturated fatty acid of linseed oil on performance and 3 health of dairy calves," addresses the timely and important topic of improving the growth and health of dairy calves through dietary supplementation. The aim of the paper is to investigate the impact of ethyl esters of polyunsaturated fatty acids on feed intake, body weight gain, feed efficiency, and health of dairy calves. The main contributions of the study include demonstrating better intake, weight gain, and health outcomes in calves supplemented with EEPUFALO. The strengths lie in the well-defined experimental design and significant results that highlight the potential benefits of the supplement.

The paper aligns well with the scope of the journal, given its focus on animal nutrition and health. However, in its current form, the manuscript has several shortcomings. The abstract could be more detailed to better reflect the study's nuances. The literature review might benefit from a more comprehensive analysis of existing research, and the methodology section could provide more details on certain aspects, such as randomization and blinding. Additionally, the authors should consider addressing potential confounding variables and economic implications of implementing EEPUFALO in calf rearing. While the conclusions are generally consistent with the evidence presented, more explicit connections between results and implications would enhance the paper's overall coherence. The references appear appropriate, but some minor adjustments could improve their alignment with the manuscript content.

In summary, the paper contributes valuable insights into improving the growth and health of dairy calves through dietary supplementation. However, refining the literature review, providing more methodological details, addressing potential confounding factors, and offering clearer connections between results and implications would strengthen the overall quality of the manuscript.

Specific comments:

·       To enhance the research's appeal, I suggest avoiding the inclusion of terms in the keywords that are already present in the article title;

·       In order not to make reading the section redundant, I recommend avoiding repetition of concepts already explained previously in the text; for example, the indication of faecal fluidity given in parentheses in relation to diarrhea;

·       Starting the Results and Discussion section by reiterating the aim of the study can provide clarity and context for readers;

·       It would be valuable to include an assessment of feed palatability in your study. Feed palatability is a critical factor influencing feed intake and, subsequently, animal performance. It can significantly affect the acceptance and consumption of specific feed components. Adding a section discussing feed palatability and citing relevant references in animal feeding practice would enhance the comprehensiveness of your study. Consider to cite: 10.1016/j.applanim.2020.105110

·       I recommend incorporating a discussion paragraph highlighting the significance of educating future veterinarians, technicians, and farmers about the issues addressed in the paper. Emphasizing the importance of effective teaching methods in shaping knowledgeable students and proficient veterinarians would add depth to the paper's implications. It is advisable to refer to recent publications on veterinary education to provide up-to-date insights into best practices in preparing future professionals to address the challenges discussed in the paper. Please see: 10.1016/j.jevs.2023.104537 and 10.3390/ani13223503.

·       The use of flaxseed as an omega 3 supplement has been adopted as an additive in the diet of dairy ewes achieving benefits on the growth performance of dairy lambs 10.3390/ani10010025, please provide a comparison and discussion of thant.

·       I kindly suggest expanding the Conclusions section of your paper to provide a more detailed and comprehensive report of the main findings. This will help readers better understand the significance of your research;

·       There are some editing issues. It's recommended to thoroughly review the document for such problems;

Author Response

(The authors gave the same response as above.)

Round 2

Reviewer 1 Report

Comments and Suggestions for Authors

The manuscript has been enhanced, addressing the concerns raised by the reviewers. However, there are still some points that need further revision:

LL 45-47: Regarding the citation, for instance, I propose citing: 10.3390/vetsci10090554.

Line 85: I fully understand the research needs; however, since the trial was conducted in European territory, I expect the rearing procedures to adhere to the guidelines established for the welfare protection of calves (Directive 2008/119/EC). Therefore, I would appreciate it if the authors could specify the company's compliance with these guidelines. Additionally, I would like to bring to the authors' attention that in Europe, it is not permissible to rear calves in individual cages beyond 8 weeks of age (i.e., beyond 56 days). Are the authors certain that the calves were consistently raised in individual cages?

L 96: The abbreviation has already been explained in the abstract section.

L 97: Is 63 days of age the commencement of the weaning period? If so, please specify.

LL 106-107: Provide the descriptive statistics of the data utilized for calf selection.

L 289: Standardize the citation style.

L 296: Standardize the citation style.

L 319: The citation for Spitalmiak-Bajerska et al. is listed with the wrong number.

L 339: It is suggested to refrain from including citations in the Conclusion section to maintain clarity.

Author Response

Thank you very much for taking the time to review this manuscript again. Please find the detailed responses in the file attached.

Reviewer 3 Report

Comments and Suggestions for Authors

The authors have diligently addressed the review comments, significantly enhancing the paper's quality. As a result, it is now well-suited for publication.

Author Response

The authors would like to thank the reviewer for his/her kind response.